# Cellular Therapy for Lung Cancer: Focusing on Chimeric Antigen Receptor T (CAR T) Cells and Tumor-Infiltrating Lymphocyte (TIL) Therapy

**DOI:** 10.3390/cancers15143733

**Published:** 2023-07-23

**Authors:** Vatsala Katiyar, Jason Chesney, Goetz Kloecker

**Affiliations:** James Graham Brown Cancer Center, University of Louisville, Louisville, KY 40202, USA

**Keywords:** NSCLC, non-small cell lung cancer, lung cancer, CAR T, chimeric antigen receptors, CAR T cell therapy, TILs, tumor infiltrating lymphocytes, TIL therapy, adoptive cell therapy, personalized medicine

## Abstract

**Simple Summary:**

Lung cancer is the leading cause of cancer related mortality and morbidity in the United States and worldwide. The advent of Immunotherapy has significantly improved lung cancer prognosis. However, there is a huge unmet need for novel agents, as a significant number of patients do not have durable responses to immunotherapy. This review article highlights two such novel techniques—Chimeric antigen receptor (CAR) T-cell therapy and Tumor-infiltrating lymphocyte (TIL) therapy. Both these techniques utilize a patient’s own immune cells to fight against tumors. While CAR T-cell therapy requires genetic modification of a patient’s T cells to express receptors that can recognize and attack tumor cells rapidly, TILs involves extraction of immune cells from tumors and their proliferation in a laboratory before being infused back to the patient. Both these techniques are currently used in a clinical trial setting only. In this review, we discuss the limitations and future directions and potential for both these treatment strategies.

**Abstract:**

Lung cancer is a leading cause of morbidity and mortality in the United States and worldwide. The introduction of immune checkpoint inhibitors has led to a marked improvement in the outcomes of lung cancer patients. Despite these advances, there is a huge unmet need for therapeutic options in patients who are not candidates for targeted or immunotherapy or those who progress after first-line treatment. With its high mutational burden, lung cancer appears to be an attractive target for novel personalized treatment approaches. In this review, we provide an overview of two adoptive cell therapy approaches–chimeric antigen receptors (CAR) T-cell therapy and Tumor-infiltrating lymphocytes (TILs) in lung cancer with an emphasis on current challenges and future perspectives. While both these therapies are still in the early phases of development in lung cancer and need more refinement, they harbor the potential to be effective treatment options for this group of patients with otherwise poor prognoses.

## 1. Introduction

Lung cancer continues to be a leading cause of mortality and morbidity worldwide. GLOBOCAN 2020 estimates of cancer incidence and mortality showed that lung cancer remained the leading cause of cancer death, with an estimated 1.8 million deaths (18%) in 2020 [1]. In the US, an estimated 238,340 people will be newly diagnosed and 127,070 people will die of lung cancer in 2023, making it the most common cause of cancer-related mortality. While advances in lung cancer treatment and the adoption of screening have led to a steady decline in overall age-adjusted lung cancer incidence rates and death rates in the US, the high mortality rates indicate a huge unmet need for effective therapeutic options [2].

Enhanced understanding of tumor microenvironment and the success of immune checkpoint inhibitors (ICIs) have transformed the treatment landscape of lung cancer. Since the approval of CTLA 4 inhibitor ipilimumab for the treatment of advanced melanoma in 2011, multiple other clinically relevant antibodies blocking PD-1 and PDL-1 have been approved for different disease states including lung cancer in both advanced and adjuvant settings.

While some patients can achieve durable responses to immunotherapy, a vast majority of patients are refractory to checkpoint blockade. The mechanisms of such primary and acquired resistance to immunotherapy can be a consequence of factors innate to the neoplastic cells as well as to the cells of the tumor microenvironment (TME). Potential intrinsic factors include alteration in signaling pathways, dedifferentiation with loss of tumor antigen expression, cancer cells genetically negative for inducible PD-L1 expression, changes in gene expression of immune-related genes due to epigenetic modification of the DNA. On the other hand, extrinsic mechanisms include low T effector to T reg cell ratio, and infiltration of Myeloid-derived suppressor cells (MSCs) and tumor-associated macrophages (TAM) [3]. Certain mutations like EGFR, ALK, KRAS, and STK11 confer poor response to ICIs [4,5,6].

There is a scarcity of effective second-line treatment options for lung cancer patients with median overall survival ranging from 8–11 months [7,8]. Various novel therapies are being explored to improve the outcomes of these patients. New immunomodulatory pathways and checkpoints are being developed. Antibodies against co-inhibitory immune checkpoints, targeting LAG3, TIM3, TIGIT, and BTLA, as well as antibodies against co-stimulatory targets, such as GITR, OX40, 41BB, and ICOS, are under development [9]. The anti-lymphocyte activation gene 3 (LAG-3) antibody, relatlimab, has been approved in combination with nivolumab for untreated metastatic melanoma [10]. A phase II clinical trial is currently evaluating relatlimab plus nivolumab in combination with chemotherapy vs. nivolumab and chemotherapy as first-line treatment for stage IV or recurrent non-small cell lung cancer (NSCLC) (NCT04623775).

Another approach that has been successful in hematologic malignancies is a bispecific T cell engager (BiTE) which is being tried in small cell lung cancer (SCLC). Tarlatamab, a half-life extended BiTE targeting delta-like protein 3 (DLL3) and CD3, has demonstrated manageable safety and durable responses in heavily pretreated patients with SCLC in a phase I trial [11]. This trial continues to enroll SCLC patients (NCT03319940).

Other important innovations include lymphokine activated killer cells, cytokine-activated killer cells, dendritic cell vaccines, tumor-infiltrating lymphocytes (TILs), and chimeric antigen receptor T cells (CAR T-cells) [12]. In this review article, we describe recent advances in cellular therapy for lung cancer with a focus on CAR T cells and TILs.

### Rationale of Use of Cellular Therapy in Lung Cancer

The success of cellular therapy in other fatal malignancies has encouraged the advancement of this treatment approach in lung cancer. While CAR T-cells have demonstrated significant efficacy and durable responses in advanced refractory hematologic malignancies including Diffuse large B cell lymphoma [13,14], other aggressive B cell lymphomas [15,16], acute lymphocytic leukemia [17], and multiple myeloma [18,19], TILs has been successfully used in metastatic melanoma [20]. Due to a high number of somatic mutations and consequent tumor neoantigens seen in lung cancers [21], it also represents an attractive target for TILs.

## 2. CAR T-Cell Therapy

Chimeric antigen receptors (CAR) are recombinant receptors, which, when engineered into the T lymphocytes, facilitate targeted antigen binding and T cell activation, thus ensuring immune attack on tumors [22]. The generation of CAR T-cells is an intricate and complex process that involves engineering T cells to express tumor-specific antigen-targeted CARs on the surface of T cells to ensure a targeted response. The process of autologous CAR T-cell manufacturing is summarized in Figure 1. CARs have evolved over time to include more costimulatory domains which have improved their efficacy and proliferation. Currently, fourth-generation CARs, also known as TRUCKS (T cells redirected for antigen-unrestricted cytokine-initiated killing) with the ability to release cytokines, upon engagement of CARs with target tissues are being developed. These CARs have been engineered to code for an array of cytokines including IL-7, IL-12, IL-15, IL-18, IL-23, and their combinations to enhance CAR cytotoxicity and efficacy [23].

Arguably, the most critical first step in the development of CARs is identifying a tumor-associated antigen (TAA) that is expressed ubiquitously and selectively on cancer cells [24]. A commonly identified mechanism of resistance of CARs in hematologic malignancies has been loss or the downregulation of the target antigen, resulting in antigen-negative and antigen-low tumor variants, after exposure to CAR T-cells [25]. Because solid tumors demonstrate greater heterogeneity in antigen expression, identifying a stable and evenly expressed TAA is an even bigger obstacle to its success in solid tumors.

The importance of identifying a stable TAA is further underscored by the suboptimal results seen from cancer vaccine trials in NSCLC. Therapeutic cancer vaccines have demonstrated disappointing clinical benefits in NSCLC due to the major challenge associated with identifying antigens that are not only abundantly expressed on tumor cells but are also identified as ‘non-self’ [26]. Unfortunately, the alteration of specific T cell responses against cancer cells and ineffective tumor infiltration by effector cells due to immunosuppressive TME has contributed to discouraging results in vaccine trials [27].

Another issue with antigen identification is the concern for on-target-off tumor-mediated side effects. The bystander effects on normal tissues due to the selection of non-specific antigens expressed in healthy cells can lead to life-threatening adverse events. A case report described the respiratory arrest and subsequent death of a metastatic colon cancer patient within 15 min of infusion of HER2/neu-specific CAR T [28]. On-target–off tumor effects have also been seen with carbonic anhydrase IX-specific CAR T-cell therapy in renal carcinoma where the bystander effect was noted on bile duct epithelium leading to cholestasis [29]. Thus, identifying the correct TAA is not only important to ensure CAR efficacy, but also its safety.

Several targets undergoing investigation for NSCLC CARs include epidermal growth factor receptor (EGFR), human epidermal growth factor receptor 2 (HER2), carcinoembryonic antigen (CEA), mesothelin (MSLN), prostate stem cell antigen (PSCA), mucin 1 (MUC1), tyrosine kinase-like orphan receptor 1 (ROR1), and programmed death ligand 1 (PD-L1) among others.

We have summarized the ongoing phase I/II trials in Table 1.

In addition to NSCLC, there has been a slow but steady advancement in development of cellular therapy in small cell lung cancer (SCLC). A potential antigen for SCLC is DLL3 that has been found to selectively overexpress on these cells [30]. DLL3-CAR NK-92 cells have shown to engage and kill DLL3 + SCLC cells efficiently and specifically in pre-clinical studies [31]. In fact, a phase I clinical trial of CAR T cell targeting DLL2 is currently ongoing (NCT03392064). In another preclinical study by Zhang et al., safety and efficacy of allogenic CAR T cells targeting DLL in SCLC was demonstrated. Allogeneic CAR T cell therapy can avoid manufacturing delays and variability in potency. These “off the shelf” CAR T cells from healthy donors may be an approach to make this treatment more readily available in the future. [32]

### 2.1. Toxicities with CAR T Cell Therapy

Experience with CAR T cell therapy in hematologic malignancies has demonstrated various toxicities including cytokine release syndrome (CRS), immune effector cell-associated neurotoxicity syndrome (ICANS), tumor lysis syndrome (TLS), on-target–off-tumor effects, anaphylaxis, and hematologic toxicities [33]. Varying grades of CRS and ICANS are most often diagnosed and have been detailed below.

Cytokine release syndrome. CRS, a potentially life-threatening toxicity associated with CAR T cell infusion, is caused by widespread activation and proliferation of lymphocytes and myeloid cells which secrete cytokines including IL-6, IFN-γ, IL-8, IL-10, granulocyte-macrophage colony-stimulating factor, and iNOS which in turn creates a state of systemic inflammation [34,35]. It is characterized by fever, hypotension, hypoxia, nausea, fatigue, and cardiac dysfunction. Treatment includes supportive care, corticosteroids, and judicious use of tocilizumab, an Il-6 antagonist [36].Immune effector cell-associated neurotoxicity syndrome. ICANS is characterized by varied neurological symptoms including but not limited to headache, aphasia, memory loss, delirium, focal weakness, and seizures [37]. Although the pathophysiology is not entirely clear, endothelial activation, and multifocal vascular disruption leading to increased blood–brain barrier (BBB) permeability have been noted in patients with severe neurotoxicity. BBB disruption and consequent elevated concentrations of systemic cytokines in cerebrospinal fluid coupled with CNS-specific production of chemokines are thought to precipitate the neurological side effects [37,38].

As outcomes of patients continue to improve with an enhanced understanding of the mechanisms and management of these adverse effects, certain challenges with the efficacy of CAR-T cells in solid tumors persist.

### 2.2. Challenges of CAR T Cell Therapy in Lung Cancer

As previously discussed, tumor microenvironment comprises of an intricate interplay between host immune cells and tumor cells which can obstruct the activity of CARs. In this section, we will discuss the potential mechanisms of resistance to CAR-cell therapy.

Antigen escape. CAR-T cells that infiltrate the tumor are known to rapidly lose their activity as tumor cells evolve after exposure [25]. The loss of target antigens has previously been seen in ALL patients where durability of CAR response is hampered by emergence of CD 19 negative leukemia [39]. Potential pathways for such antigen escape include selection of cells with alternative target expression that lacks the binding site for CARs or lineage switching and phenotypic evolution of cancer cells [40,41]. In addition to hematological malignancies, this phenomenon of initial response and later resistance has also been seen in glioblastoma treated with intracranial CAR T cell therapy targeting IL13Rα2 and it has been hypothesized that decreased tumor burden and immune rejection of CAR T product could be responsible [42]. Thus, identifying a homogenously and steadily expressed target antigen is of utmost importance.Tumor heterogeneity. Cancers are dynamic and genomically unstable with spatial and temporal heterogeneity. While spatial heterogeneity refers to unequal distribution of genetically distinct tumor subpopulations across different disease sites or within a single disease site, temporal heterogeneity implies the evolution of tumors over time under different selection pressures [43]. The spatial heterogeneity and its impact on survival have been successfully demonstrated in lung cancer patients [44]. The ability of CARs to recognize a singular target in a constantly changing microenvironment that is also spatially diverse reduces its activity.Immunosuppressive TME, physical barrier, and T cell exhaustion. The activity of CAR T-cells is further impeded by immune response suppressive cells including MSCs, cancer-associated fibroblasts, TAMs, and regulatory T cells in TME. The stromal cells along with tumor cells release a host of immunosuppressive cytokines including TGF-β, IL-10, ARG-1, inducible nitric oxide synthase, COX2, PGE2, FAP, and PD-L1 to help the tumor evade CAR T cells [3,45]. Additionally, effective infiltration of T cells into the TME is affected by the density of the stromal extracellular matrix (ECM) with poor migration seen in areas of dense ECM [46]. Even after successful infiltration and engagement, the durability of CAR T response is downregulated by T cell exhaustion mediated by chronic antigen stimulation and upregulation of the NR4A transcription factor family [47].

### 2.3. Future Perspectives

To mitigate the unique challenges of CAR T-cell therapy in solid tumors, genetic engineering techniques are being utilized to modify their structure to ensure persistent and durable efficacy.

Multi-specific CARs that can engage with multiple tumor antigens are being developed to overcome problems associated with antigen escape and antigen loss. Strategies for this include split universal and programmable (SUPRA) CAR and leucine-zipper motif CAR (ZipCAR). Due to their high specificity, they are also able to decrease on-target-off-tumor effects [48]. Small molecules-based or chemogenetic-based switchable CAR T-cells have been developed to regulate CAR activity and have shown increased efficacy towards cancer cells [49,50]. The development of the fourth-generation CARs called TRUCKs, which not only have direct cytotoxic effects but are also able to modulate TME by releasing cytokines, can potentially increase their efficacy as well as specificity [19].

Improving the metabolic properties of CARs so that they are not inhibited by hypoxia, reactive oxygen species, and suppressive effects of other toxic metabolites would also increase CAR efficacy [51]. These modifications to CAR structure would not only help ensure effective CAR trafficking to tumor site but also circumvent T cell exhaustion.

In addition to improving the structure and properties of CARs by inducing multiantigen specificity and cytokine release amongst others, combining CAR T cells with other traditional anticancer therapies and immunotherapy also produce synergistic effects and hence improved activity [52].

## 3. Tumor-Infiltrating Lymphocyte Therapy

Tumor stroma is composed of a multitude of cells including T cells, B lymphocytes, macrophages, natural killer cells, and dendritic cells, collectively known as TILs [53]. This admixture of cells predominantly consists of polyclonal, mostly tumor-specific CD8+ and CD4+ T cells which can bind to multiple TAAs and overcome the tumor heterogeneity and antigen escape [54]. Due to the immunosuppressive TME, these cells lose their ability to attack the tumor, however when extracted from the tumor and expanded ex vivo, they get reactivated to target the neoplastic cells [55].

The utility of autologous TILs therapy as an effective anti-cancer treatment was first demonstrated in metastatic melanoma patients by Rosenberg and colleagues. They harvested TILs from the patient’s own tumor and expanded them in vitro. The process of in vitro T cell expansion is demonstrated in Figure 2. The cryopreserved cells were infused back into the lymphodepleted patients. The patients then received weight-based IL-2 immune modulation for several doses in the intensive care setting. Regression of tumors was noted in 11 of 20 treated patients in this trial which was higher than objective response rates achieved with IL-2 or lymphokine-activated killer cells administered alone [56]. This heralded the era of development and optimization of TILs therapy in other cancer types including ovarian, lung, cervical, and breast cancer [57,58,59,60].

Despite the high clonal mutation burden and a high number of tumors neoantigens in NSCLC [21] that could predict a favorable response to immunotherapy [61], presumably due to the development of neoantigen-specific T cells that direct anti-tumor immunity [62], the early trials of TILs in lung cancer yielded dismal responses. In the first clinical trial of TILs in lung cancer patients by Kradin et al., none of the patients achieved greater than 50% reduction of total tumor burden [58]. Similarly, poor response rates were seen in another trial involving NSCLC patients [63]. Ratto et al. tested TILs in post-operative setting in stage II and III NSCLC, and found that 18 out of 113 cultures did not yield any growth. The remaining patients were randomized to receive standard treatment vs. TILs. Three-year survival was greater for patients who received TILs. While stage II patients did not benefit, it did help stage IIIB patients, which demonstrated the potential feasibility of TILs in NSCLC [64].

Improved understanding of TME and tumor biology including the immunological make-up of tumors, immune checkpoints, and immune exhaustion has made this form of treatment a potential breakthrough for a fatal disease that has limited second-line therapy options [54]. It is still unclear which of the lymphocyte subtypes is the driving force for the effectiveness of TIL therapy, but it appears to be related to neo-antigen reactive T cells within the TME. It has been demonstrated that while higher levels of CD8+, CD4+, and CD3+ T cells in tumor stroma are associated with better overall survival (OS), FOXP3+ T cell infiltration leads to decreased OS in NSCLC patients [65,66]. Additionally, an interplay between infiltrating CD4+ T cells and CD8+ T cells in tumors might be more important in the suppression of the progression of NSCLC than their isolated presence alone, as CD4+ T cells play an important role by secreting cytokines such as IL-2, which promotes CD8+ T cell growth and proliferation [66]. On the other hand, the presence of T regulatory cells leads to the propagation of immunosuppressive TME and has been associated with tumor growth and metastasis [67,68].

### 3.1. Recent Clinical Trials

With this improved understanding of lung tumor immunobiology and advances in technology, Avi and colleagues explored the feasibility of TIL therapy in NSCLC patients. They were able to isolate multiple TIL cultures from five NSCLC patients and successfully expand them. The results of this pre-clinical evaluation established the feasibility of TILs in a population with unmet need [69].

In a recent phase I single-arm clinical trial, Creelan et al. tested the safety and efficacy of TILS on pretreated NSCLC patients. This trial included patients with EGFR or ALK translocations if they had progressed on ≥1 previous approved TKIs. Those with active brain metastasis and who had received immunotherapy prior to clinical trial were excluded. PD-(L)1 blockade naïve patients were selected to reduce the proportion of terminally differentiated T cells in culture.

After tumor harvesting, patients received at least four cycles of nivolumab. If there was tumor progression on two sequential scans, they proceeded with cyclophosphamide and fludarabine lymphodepletion. Patients then received autologous TILs that had been expanded ex vivo with IL-2, followed by interleukin-2 infusions. This was followed by maintenance of nivolumab for a year. The primary endpoint was safety and secondary endpoints included objective response rate (ORR), duration of response, and T cell persistence. Out of 20 patients included in the trial, 16 received TIL therapy. Thirteen patients were evaluated for a response. Radiographic response, including unconfirmed response, occurred for 6 of 13 evaluable patients. These included two complete responses which remained ongoing 1.5 years later. Overall, the median best change in the sum of target lesion diameters was −35.5% (range +20 to −100).

Common nonhematologic adverse events included hypoalbuminemia, hypophosphatemia, nausea, hyponatremia, and diarrhea. Two patients died before response assessment. Neutrophil count recovered in a median duration of 7.5 days. The majority of TIL-related adverse events resolved within one month of infusion. This study highlighted that TILs could be successfully harvested and expanded from NSCLC patients and can help achieve durable responses in patients who are otherwise nonresponsive to immunotherapy. However, this approach needs to be tested in wider patient populations and real-world scenarios to assess risk and cost–benefit ratios [55].

Another study led by Schoenfeld et al. reported the first safety and efficacy data for single-agent LN-145 TIL cell therapy in patients with advanced NSCLC. It was a multicenter, multicohort, open-label phase-two study. Cohort 3B included advanced NSCLC patients who had been treated with 1–3 prior lines of systemic therapy including either ICI or oncogene-directed therapy. Unlike the trial by Creelan et al., all patients had previously received immunotherapy.

Primary endpoints included efficacy as defined by ORR and safety as measured by incidence of grade ≥3 treatment-emergent adverse effects (TEAE, defined as adverse events that occur from the time of TIL infusion, up to 30 days after TIL infusion or the start of a new anticancer therapy).

Out of a total of 28 NSCLC patients who received TILs in this trial, 24 patients had ≥1 efficacy assessment. ORR was noted in 6 patients (25%), 12 had stable disease and 6 patients had disease progression in the efficacy evaluable set. One patient had a complete metabolic response, ongoing at 20.7 months. The safety profile was consistent with the known safety profiles of non-myeloablative regimens and IL2 infusions with bone marrow suppression, hypotension, hypoxia, and fatigue being the most common TEAEs. This signal-finding study demonstrated that TILs could be a potential earlier line treatment option in NSCLC patients who had previously received immune checkpoint inhibitors [70]. Other ongoing phase I and II trials of TILs in NSCLC are summarized in Table 2.

### 3.2. Challenges of TIL Therapy in Lung Cancer

Despite the success of TILs in these earlier phase studies, its development and use in real-world scenarios is still in its infancy. Amongst various challenges associated with this treatment, include successful isolation and selected expansion of TILs which can take up to 6–8 weeks. Patients with progressive disease can develop an overwhelming symptom burden during that time frame which can disqualify them from receiving the treatment [54].

Furthermore, the inability to efficiently identify and isolate neoantigen-specific lymphocytes and the presence of immunosuppressive cells in TME could further impede their anti-tumor activity by causing cytotoxic T cells exhaustion. Additionally, the infused T cells often have a short half-life in vivo and may not be able to effectively migrate to target tumor locations, leading to suboptimal efficacy [71].

While this form of personalized cellular therapy has fewer on-target-off-tumor effects as compared to CAR T, it does have unique adverse effects. TILs infusion is preceded by non-myeloablative chemotherapy regimens and is followed by IL-2 infusions. This requires a prolonged hospitalization and various TEAEs including bone marrow suppression, hypoxia, hypotension, diarrhea, and fevers can occur. Furthermore, due to the toxicity associated with this treatment, TILs may not be appropriate for all metastatic NSCLC patients and a careful selection of suitable patients with good cardiopulmonary reserve is quintessential for the success of this therapy [55,70].

### 3.3. Future Perspectives for TIL Therapy

Efforts are currently ongoing to enhance the efficacy of TILs by genetically manipulating peripheral blood lymphocytes for autologous cellular therapy protocols. Most of these genetic engineering approaches involve the removal of programmed cell death protein 1 (PD-1, PDCD1) from T cells, thus precluding its interaction with programmed death ligand 1 (PD-L1) on antigen-presenting and tumor cells. The manipulation of the PD-1/PD-L1 axis could not only enhance the efficacy of TILs but also eliminate the immunotherapy adverse events seen because of T cells being affected at non-tumor locations. Common gene editing techniques being used include CRISPR-Cas9, Zinc finger nucleases (ZFNs), and transcription activator-like effector nucleases (TALENs) [72]. A first in-human phase 1/2 trial of TALEN-mediated PD-1–inactivated TILs (PDCD-1 knockout TILs) is currently ongoing with a plan to enroll 53 patients with metastatic melanoma and NSCLC to assess the safety and ORR [73].

Previously, in vitro studies had shown that genetically engineering TILs to express IL-2 could prolong their survival after IL-2 withdrawal while maintaining their tumor specificity and function; however, this approach did not yield effective anti-tumor response or lead to prolonged TIL survival in vivo in melanoma patients [74]. Other approaches to enhance the efficacy of TILs have included the introduction of the CXCR2 gene into tumor-specific T cells to enhance their tumor-specific migration, localization, and in turn anti-tumor response [75] as well as T cells genetically modified to resist exogenous TGF-β signaling, which is known to inhibit tumor-specific cellular immunity [76]. While gene editing efforts in TILs are ongoing, the efficacy data in pre-clinical and clinical settings are awaited.

Additionally, the development of more standardized, rapid, and cost-effective TILs would help expand the availability of this novel therapy from limited premier research institutes to the community. Moreover, TILs are being combined with chemo/radiotherapy, immunotherapy, and oncolytic virotherapy in other cancer types, and such combination approaches may be further explored in NSCLC as well [54].

## 4. Conclusions

Advances in the understanding of tumor immunobiology coupled with the development of sophisticated genetic engineering techniques have led to major advances in personalized cellular therapy for lung cancer patients. Novel strategies to identify key tumor neoantigens, overcome T cell exhaustion, reduce the processing time of T cells, combat treatment-related toxicity, and ensure widespread availability will ultimately predict the success of these therapies. Development of allogeneic CAR T cells and next-generation TILs can help overcome some of these concerns. More pre-clinical and clinical studies are needed to further discern the appropriate clinical use of these novel therapies.

## Figures and Tables

**Figure 1 cancers-15-03733-f001:**
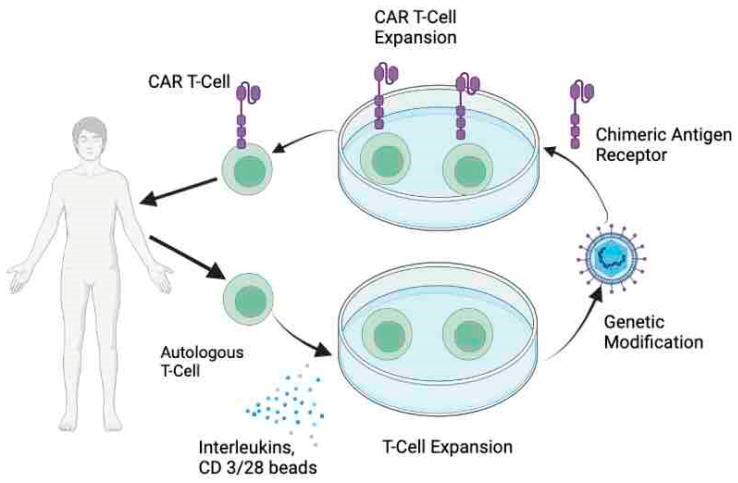
Autologous CAR T cell production.

**Figure 2 cancers-15-03733-f002:**
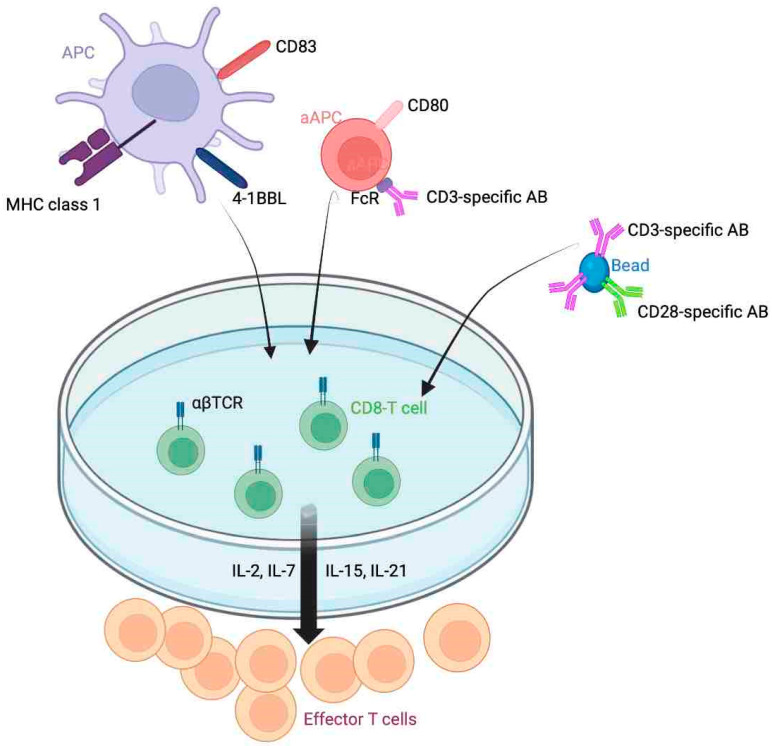
T cell expansion in vitro. CD8+ T cells expanded by antigen-presenting cells (APC) and/or beads with stimulatory antibodies (CD3, CD28) or artificial aAPCs with stimulatory ligands (CD83, 4-1BBL) in the presence of IL cytokines.

**Table 1 cancers-15-03733-t001:** Summary of active and recruiting clinical trials of CAR T cell therapy in lung cancer (Available at https://clinicaltrials.gov, accessed on 1 July 2023).

Trial Identifier	Title	Phase	Types of Cancers
NCT05341492	EGFR/B7H3 CAR T on lung cancer lung cancer and triple negative breast cancer	I	EGFR-B7H3 Positive cancers
NCT03198052	GPC3/mesothelin/claudin18.2/GUCY2C/B7-H3/PSCA/PSMA/MUC1/TGFβ/HER2/Lewis-Y/AXL/EGFR-CAR T cells against cancers	I	Lung cancer
NCT05060796	Study of CXCR5 modified EGFR targeted CAR T cells for advanced NSCLC	I	NSCLC
NCT05239143	P-MUC1C-ALLO1 allogeneic CAR T cells in the treatment of subjects with advanced or metastatic solid tumors	I	NSCLC and multiple cancers
NCT05483491	KK-LC-1 TCR-T cell therapy for gastric, breast, cervical, and lung cancer	I	Lung cancer and multiple cancers
NCT05274451	A study to investigate LYL797 in adults with solid tumors	I	Breast cancer Lung cancer
NCT05120271	BOXR1030 T cells in subjects with advanced GPC3-positive solid tumors	I	Squamous NSCLC
NCT03740256	Binary oncolytic adenovirus in combination with HER2-specific autologous CAR T VST, advanced HER2 positive solid tumors (VISTA)	I	HER 2 positive lung and multiple cancers
NCT05736731	A study to evaluate the safety and efficacy of A2B530, a logic-gated CAR T, in subjects with solid tumors that express CEA and have lost HLA-A*02 expression (EVEREST-1)	I/II	NSCLC and multiple cancers
NCT04348643	Safety and efficacy of CEA-targeted CAR T therapy for relapsed/refractory CEA+ cancer	I/II	NSCLC and multiple cancers
NCT03198546	GPC3-CAR T cells for immunotherapy of cancer with GPC3 expression	I	Squamous cell lung cancer, HCC
NCT02414269	Malignant pleural disease treated with autologous T cells genetically engineered to target the cancer-cell surface antigen mesothelin	I/II	NSCLC and multiple cancers

**Table 2 cancers-15-03733-t002:** Summary of active and recruiting clinical studies of TILs therapy in lung cancer (Available at https://clinicaltrials.gov, accessed on 1 July 2023).

Trial Identifier	Title	Phase	Types of Cancer
NCT04614103	Autologous LN-145 in patients with metastatic non-small-cell lung cancer	II	NSCLC
NCT05681780	Clinical trial of CD40L-augmented TIL for patients with EGFR, ALK, ROS1, or HER2-Driven NSCLC	I/II	NSCLC
NCT02133196	T cell receptor immunotherapy for patients with metastatic non-small cell lung cancer	II	NSCLC
NCT05366478	A clinical study of LM103 injection in the treatment of advanced solid tumors	I	NSCLC Cervical cancer Melanoma
NCT03645928	Study of autologous tumors infiltrating lymphocytes in patients with solid tumors	II	Lung Head and neck Melanoma
NCT05361174	A study to investigate the efficacy and safety of an infusion of IOV-4001 in adult participants with unresectable or metastatic melanoma or stage III or IV non-small-cell lung cancer	I/II	Melanoma NSCLC
NCT05573035	A study to investigate LYL845 in adults with solid tumors	I	Melanoma Lung cancer Colorectal cancer
NCT03778814	TCR-T cell immunotherapy of lung cancer and other solid tumors	I	NSCLC Solid tumors
NCT03215810	Nivolumab and tumor infiltrating lymphocytes (TIL) in advanced non-small cell lung cancer	I	NSCLC
NCT05878028	L-TIL plus tislelizumab for PD1 antibody resistant aNSCLC	II	NSCLC
NCT05397093	ITIL-306 in advanced solid tumors	I	NSCLC Epithelial ovarian and Renal cell carcinoma
NCT05576077	A study of TBio-4101 (TIL) and pembrolizumab in patients with advanced solid tumors (STARLING)	I	NSCLC and multiple other cancers
NCT05680922	DLL3-directed chimeric antigen receptor T cells in subjects with extensive stage small cell lung cancer	I	SCLC and large cell neuroendocrine carcinoma of lung

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
