# Peer review of "Cellular Therapy for Lung Cancer: Focusing on Chimeric Antigen Receptor T (CAR T) Cells and Tumor-Infiltrating Lymphocyte (TIL) Therapy"

_cancers, 2023, doi:10.3390/cancers15143733_

Round 1

Reviewer 1 Report

Katiyar et al submitted a Review titled "Cellular therapy in lung cancer" to be considered for Cancers. The authors focused on two recent approaches based on CAR-T cell therapy and TILs for lung cancer. Although the topic is interesting, this Review requires MAJOR REVISIONS before to be considered for publication. 

- The title must revised. Indeed, the authors are focusing on only two approaches. This title is something misleading. 

- Introduction section must be deepened. The authors should discuss about immunotherapy for cancer in general in this section. You can consider the following articles:

Int. J. Mol. Sci. 202324(4), 3086; https://doi.org/10.3390/ijms24043086

Nat Rev Clin Oncol 2022, 19, 37–50. https://doi.org/10.1038/s41571-021-00552-7

Exp Hematol Oncol 2023, 12, 5. https://doi.org/10.1186/s40164-022-00368-w

- Recent studies are evaluating safety, tolerability and efficacy of BiTe for lung cancer (NCT03319940). I suggest to add a new section.

English language and style are fine/minor spell check required. 

Author Response

- The title must be revised. Indeed, the authors are focusing on only two approaches. This title is something misleading. 

We thank the reviewer for this very useful comment. We have changed the title of the article to “Cellular Therapy for lung cancer:  Focusing on Chimeric Antigen Receptor T (CAR-T) cells and Tumor-infiltrating lymphocyte (TIL) therapy” to reflect the two approaches described in detail in this review.

- Introduction section must be deepened. The authors should discuss about immunotherapy for cancer in general in this section. You can consider the following articles:

Int. J. Mol. Sci. 202324(4), 3086; https://doi.org/10.3390/ijms24043086

Nat Rev Clin Oncol 2022, 19, 37–50. https://doi.org/10.1038/s41571-021-00552-7

Exp Hematol Oncol 2023, 12, 5. https://doi.org/10.1186/s40164-022-00368-w

We thank the reviewer for their insight. We have modified the introduction to reflect addition of information from the following article: Nat Rev Clin Oncol 2022, 19, 37–50. https://doi.org/10.1038/s41571-021-00552-7. We have focused more on mechanism of immune checkpoint inhibitor resistance, as we want to justify the need of TILs and CAR-T cell therapy in the current immunotherapy era.

- Recent studies are evaluating safety, tolerability and efficacy of BiTe for lung cancer (NCT03319940). I suggest adding a new section.

We thank the reviewer for their suggestion. Since, BiTe therapy is beyond the scope of this review article, we have included it at as part of the introduction section.

Reviewer 2 Report

The review article titled “Cellular therapy in lung cancer” by Vatsala Katiyar et al., described well. In cellular therapy, a patients own immune cells are collected and modified to recognize and attack cancer cells. The modified cells are then injected back into the patient, where they can seek out and destroy cancer cells. Lung cancer is a serious disease that affects people all over the world. Current treatment on CAR T therapies is still being studied and improved to help even more people. The author focused on CAR T cell therapy and TILs are two new therapies being tested to help treat people with lung cancer. These therapies are still being developed for people with lung cancer in the future. I would suggest the article needs few minor corrections before considering the next level.

Minor changes:

 1.The author highlighted few ongoing clinical trials. It would be better if author can include most of the studies (For eg. NCT03330834, NCT03525782, NCT03740256, NCT02713984 etc)

2.     Table 2 needs to be updated too.

3.     What are the disadvantages of CAR T cell therapy in lung cancer. The author mentioned only one common reference

4.     Is there link between TIL and lymphodepletion in lung cancer?

Author Response

 1.The author highlighted few ongoing clinical trials. It would be better if author can include most of the studies (For eg. NCT03330834, NCT03525782, NCT03740256, NCT02713984 etc)

We thank the reviewer for this very helpful comment. We have updated Table 1 to provide information on all active and recruiting trials for CAR-T cell therapy in lung cancer.

  1. Table 2 needs to be updated too.

We appreciate reviewer’s comment. We have updated Table 2 and added all active and recruiting trials for TIL therapy in lung cancer.

  1. What are the disadvantages of CAR T cell therapy in lung cancer. The author mentioned only one common reference.

We have added a new section 2.1 titled “Toxicities with CAR-T cell therapy” in the review article in line with reviewer’s suggestions.

  1. Is there link between TIL and lymphodepletion in lung cancer?

We thank the reviewer for this insightful comment. Unfortunately, we were unable to find a link between TIL and lymphodepletion specifically in lung cancer patients on our literature review.

Reviewer 3 Report

This is a comprehensive overview on chimeric antigen receptors (CAR) T cell therapy and Tumor infiltrat-9 ing lymphocytes (TILs) in lung cancer. In general, it is well written.

Few minor aspects:

The authors correctly state that identifying tumor associated antigen (TAA) is critical. Could the elaborate a little more? Why have several vaccination studies been negative? What about bystander effects?

How to circumvent T cell exhaustion?

What about the balance between even stronger activation of the immune system versus IO toxicity?

If one had to choose between CAR and TIL strategies what would be each positives and negatives? (From the clinical view, this has been nicely described for TILs).

Author Response

The authors correctly state that identifying tumor associated antigen (TAA) is critical. Could the elaborate a little more? Why have several vaccination studies been negative? What about bystander effects?

We thank the reviewer for this useful comment to help improve the content of the review article. We have elaborated on the importance of identifying TAA in Section 2 and this has been explained further in Section 2.2, point 1 – Antigen Escape. Reason for failure of tumor vaccines and importance of bystander effects has been added and elaborated in section 2. 

How to circumvent T cell exhaustion?

We appreciate reviewer for raising attention to the important fact. We have described advancements to increase CART cell efficacy and circumvent T cell exhaustion in section 2.3- Future perspectives.

What about the balance between even stronger activation of the immune system versus IO toxicity?

We thank the reviewer for bringing up this important point. Stronger activation and combination approaches could potentially lead to more IO related toxicity. However, with the emergence of new immune check point pathways and drugs targeting them, the toxicities may potentially be reduced (as seen in relatlimab and nivolumab combination). Introduction section has been modified in response to this comment.

If one had to choose between CAR and TIL strategies what would be each positive and negatives? (From the clinical view, this has been nicely described for TILs).

We thank the reviewer for this useful comment to help improve the flow of information. We have updated the review article to include section 2.1 to highlight the toxicities associated with CAR- T cells. Both CAR-T cells and TILs provide an additional line of therapy for lung cancer patients and are in early phase trials and both have unique challenges which have been detailed in section 2.2 and 3.2.